# Synonymous Dinucleotide Usage: A Codon-Aware Metric for Quantifying Dinucleotide Representation in Viruses

**DOI:** 10.3390/v12040462

**Published:** 2020-04-20

**Authors:** Spyros Lytras, Joseph Hughes

**Affiliations:** MRC—University of Glasgow Centre for Virus Research, Glasgow G61 1QH, UK

**Keywords:** dinucleotides, CpG suppression, *Flaviviridae*, *Rhabdoviridae*, synonymous codon usage, bioinformatics, python package

## Abstract

Distinct patterns of dinucleotide representation, such as CpG and UpA suppression, are characteristic of certain viral genomes. Recent research has uncovered vertebrate immune mechanisms that select against specific dinucleotides in targeted viruses. This evidence highlights the importance of systematically examining the dinucleotide composition of viral genomes. We have developed a novel metric, called synonymous dinucleotide usage (SDU), for quantifying dinucleotide representation in coding sequences. Our method compares the abundance of a given dinucleotide to the null hypothesis of equal synonymous codon usage in the sequence. We present a Python3 package, *DinuQ*, for calculating SDU and other relevant metrics. We have applied this method on two sets of invertebrate- and vertebrate-specific flaviviruses and rhabdoviruses. The SDU shows that the vertebrate viruses exhibit consistently greater under-representation of CpG dinucleotides in all three codon positions in both datasets. In comparison to existing metrics for dinucleotide quantification, the SDU allows for a statistical interpretation of its values by comparing it to a null expectation based on the codon table. Here we apply the method to viruses, but coding sequences of other living organisms can be analysed in the same way.

## 1. Introduction

Certain dinucleotides, two nucleotides adjacent in a sequence, are known to be over- or under-represented in the genomes of living organisms, creating distinct compositional patterns [1]. Organisms with methylated genomes such as vertebrates and plants have low levels of CpG dinucleotides. This is not the case for methylase-absent organisms like invertebrates, bacteria and fungi [2,3]. The cause of CpG suppression lies in the DNA methylation mechanisms of vertebrates and plants, where cytosine is frequently converted to thymine by DNA methyltransferases [4,5]. UpA deprivation is consistently present in most living organisms, including prokaryotes. This bias is suspected to be due to UpA-rich mRNA being unstable and more prone to degradation by cytoplasmic RNAses [6,7].

Similar patterns of dinucleotide composition have been observed in RNA and DNA viruses, and appear to have a functional role in the infection and propagation of the viruses [8]. Studies have shown that experimentally increasing UpA and CpG levels in RNA viruses leads to a decrease in replication and subsequent viral attenuation [9,10,11]. Influenza CpG-/UpA-rich sequences have also been shown to have reduced replication in vivo, causing a more powerful immune response, and in the case of CpG increase, showing reduced clinical severity [12]. A similar decrease in replication and virulence has been observed in CpG-/UpA-increased yellow fever virus [13].

Recently, a host immune mechanism of vertebrates which acts on viral genomes on the dinucleotide level, has been uncovered. The zinc-finger antiviral protein (ZAP) appears to be integral to this system by targeting CpG-rich infecting viral RNA [14]. This host-driven force selects against viruses rich in CpG dinucleotides and drives the observed under-representation of CpG in HIV-1 [14], the model RNA virus Echovirus 7 [15], and even DNA viruses [16]. Additionally, UpA is mildly under-represented in most viral genomes [17]. Even though our understanding of this dinucleotide’s depletion is limited, there is some evidence suggesting that a similar antiviral mechanism might also be responsible for this pattern [15,18].

The genetic code is degenerate, meaning that some codons encode the same amino acid. These codons are called synonymous, because changes between them do not alter the resulting peptide sequence. The frequency in which synonymous codons are found in a coding sequence can be biased, as a result of various mutational or selective pressures acting on the genome [19]. Metrics for measuring codon usage bias exist, such as the relative synonymous codon usage (RSCU), which quantify the bias in synonymous codon frequencies [20]. However, these measures do not inform on the underlying dinucleotide bias in the sequence. If such forces are acting on the dinucleotide and not the peptide level in viral genomes, adaptation towards, for example, a less CpG-rich genome, would preferentially take place through synonymous changes in the sequence. This would lead to a genome with a reduced abundance of dinucleotide patterns recognized by host immunity, whilst still maintaining the protein sequence. Thus, the dinucleotide-acting compositional bias would produce a skew in the frequencies of synonymous codon usage in the genome.

Previous research has tried to disentangle dinucleotide patterns from codon and amino acid frequencies, mainly focusing on quantifying CpG patterns [21]. More complex mathematical analyses have also been implemented for understanding evolutionary change in viral genome nucleotide motifs [22]. The only method, however, that is being routinely used for quantifying dinucleotide representation is the relative dinucleotide abundance (RDA). This is simply the odds ratio between the frequency of a dinucleotide XY and the product of the frequencies of each single nucleotide X and Y across a whole sequence [23]. The evidence of a host mechanism selecting against dinucleotides in viruses highlights the importance of developing a method for systematically quantifying dinucleotide representation by comparing the observed patterns to a biologically relevant null hypothesis. We propose the synonymous dinucleotide usage (SDU), a metric that compares the occurrences of a given dinucleotide to the null hypothesis where there is equal usage of synonymous codons. This method provides a way of measuring the extent to which a host-driven force acting on the dinucleotide level of viral genomes has skewed the synonymous codon usage of the protein sequence and can be applied in all dinucleotide combinations and different sequence frame positions.

## 2. Materials and Methods

### 2.1. Synonymous Dinucleotide Usage (SDU)

A coding sequence can have three distinct dinucleotide frame positions. We define the dinucleotide frame position 1 as the first and second nucleotide position of a codon, dinucleotide frame position 2 as the second and third nucleotide position of a codon, and dinucleotide bridge position as the third nucleotide position of a codon and the first position of the downstream codon. Each one of these positions can take a set of different dinucleotides without changing the amino acid (positions 1 and 2) or amino acid pair (bridge position) in the protein sequence, we define these as a set of synonymous dinucleotides. For example, threonine has four synonymous codons ACU, ACC, ACA, ACG. In dinucleotide position 2 of a codon encoding for threonine, there are four synonymous dinucleotides, CU, CC, CA, and CG. As such, the expected proportion of CpU occurring in position 2 coding for threonine under the null hypothesis of equal synonymous codon usage is: e_i_ = 0.25. The SDU (Table 1) compares the observed proportion of a synonymous dinucleotide of interest (o_i_) to that expected under equal synonymous codon usage (e_i_) for a given dinucleotide frame position. The ratio between o_i_ and e_i_ is calculated for each different amino acid or amino acid pair (for the bridge position) and the SDU is defined as the weighted arithmetic mean of the ratios, weighted by the abundance of each amino acid in the sequence (Equation (1)).
(1)$${\mathrm{SDU}}_{j,h}=\frac{\sum_{i=1}^{k}n_{i}\times \frac{o_{i,j,h}}{e_{i,j,h}}}{N}$$

The set *j* includes 16 possible dinucleotide combinations. With three frame positions *h*, the matrix of SDU_j,h_ has 48 possible combinations. Only 3 amino acids can be encoded by different position 1 dinucleotides (arginine, serine, and leucine), meaning that 11 out of 16 dinucleotides in frame position 1 are non-informative, leaving 37 informative combinations. The result of the SDU directly reflects the overall synonymous dinucleotide representation in each frame position of a given sequence:An SDU of 1 indicates that the representation of the dinucleotide of interest in the given frame position is equal to that expected under the null hypothesis of equal synonymous codon usage;an SDU of 0 indicates that the dinucleotide of interest is completely absent in the given frame position across the sequence;an SDU greater than 1 indicates that the dinucleotide of interest is over-represented in the given frame position, compared to the representation expected under the null hypothesis;an SDU between 0 and 1 indicates that the dinucleotide of interest is under-represented in the given frame position, compared to the representation expected under the null hypothesis.

### 2.2. Relative Synonymous Dinucleotide Usage (RSDU)

The number of amino acids or amino acid pairs that can be synonymously produced by a certain dinucleotide varies between dinucleotides and frame positions. This means that SDU measurements for different positions and dinucleotides can reach different maximum values. Under- (SDU < 1) and over-representation (SDU > 1) can still be consistently interpreted between positions and dinucleotides, since an SDU of 1 always reflects complete agreement with the null hypothesis. However, the magnitude of over-representation cannot be compared between different positions and dinucleotides. In order to make this comparison possible we extended the SDU to the relative synonymous dinucleotide usage (RSDU). This is essentially the calculated SDU value, normalised by the maximum SDU for this position and dinucleotide (Equation (2)). This extension to the metric does not have a consistent scale for comparison to the null hypothesis, and instead allows for comparing the extent of the difference from the null hypothesis between the different parameters.
(2)$${\mathrm{RSDU}}_{j,h}=\frac{\sum_{i=1}^{k}n_{i}\times \frac{o_{i,j,h}}{e_{i,j,h}}}{\sum_{i=1}^{k}n_{i}\times \frac{1}{e_{i,j,h}}}$$

The results of the RSDU are as follows:An RSDU of 1 indicates that only the dinucleotide of interest is being used in the sequence, all the other synonymous dinucleotides being absent for the given position;an RSDU of 0, similar to the SDU, indicates that the dinucleotide of interest is completely absent in the given frame position.

### 2.3. Measuring the Null Distribution

The variance of the metric is expected to vary depending on the sequence length and amino acid composition. Thus, we have developed a method of measuring the null distribution by randomly populating the given sequence with synonymous codons without changing the amino acid sequence. This process takes place for a number of iterations, calculating the SDU (or RSDU) for each model sequence. This produces a distribution that represents the random error of the metric for the given sequence abiding to the null hypothesis.

### 2.4. DinuQ Python Package

We have developed a Python3 package called *DinuQ* (dinucleotide quantification) that includes modules for calculating the SDU and RSDU with the respective error distribution for any set of coding sequences provided in a FASTA file. The package further includes modules for calculating the RSCU [20] and relative dinucleotide abundance (RDA, sometimes called DRA) [23], so that the user can make comparisons between metrics. The calculated values can be provided in a Python dictionary to manipulate within the Python framework, or simply exported in a table format. The Biopython package [24] is used by the tool to facilitate sequence input. The package can be installed through the Python Package Index (PyPI). Full documentation and code are available online at https://github.com/spyros-lytras/dinuq.

### 2.5. Applying the SDU on Viral Sequences

To test the uses of the SDU we have collected polyprotein coding sequences of vertebrate- and invertebrate-specific viruses of the *Flaviviridae* family, analysed in previous research [25]. Our analysis focuses on two representatives of the dataset: The insect-specific *Aedes flavivirus* (AEFV, Genbank accession: AB488408.1) that usually affects *Aedes* spp. mosquitoes [26], and the *Apoi virus* (APOIV, Genbank accession: AF160193.1) that has no known insect vector and affects rodents of the *Apodemus* genus [27]. To extend the analysis, we assembled a second dataset by collecting the coding sequences of all members of the *Rhabdoviridae* family included in the ICTV Virus Metadata Resource (version November 27, 2019; MSL34) [28], being labelled as having a vertebrate or invertebrate host. The SDU, RSDU, RDA and RSCU values were calculated using the DinuQ Python package. General Linear Models (GLM) for statistical comparisons were performed using the R coding language.

## 3. Results

### 3.1. Effect of Sequence Length on SDU Error

Because the SDU splits up the sequence information into many categories (i.e., independent proportions of the dinucleotide of interest for each amino acid or amino acid pair), if the given sequence is short there will be less information in each category and subsequently more variability/error in the calculated SDU value. We have randomly simulated 10 amino acid sequences of different lengths sequentially increasing by 10% of the longest sequence’s length (i.e., from 700 to 7000 amino acids long). We then implemented our method of measuring error and calculated the SDU for each random sample. Figure 1a,b illustrate how the error distributions and their standard deviations vary between sequences of different lengths. By principle, the standard deviation of the error distribution should approach 0 as the sequence length increases. Our simulation experiment, however, shows that the magnitude of error is consistently very low for sequences longer than about 17,000 bp (standard deviation < 0.05).

### 3.2. Metric Comparisons Using an Insect- and a Vertebrate-Specific Flavivirus

Previous analysis [25] showed that the host environment differently affects the nucleotide, codon and dinucleotide composition in viruses of the *Flaviviridae* family. In particular, APOIV and other flaviviruses with no known insect vector show an under-representation of CpG, possibly a result of the ZAP-related vertebrate immune response targeting CpG-rich viruses. In contrast, AEFV and other insect-specific flaviviruses do not exhibit this bias. In Figure 2b and Figure 3b, we present the SDU patterns for all dinucleotides and frame positions of APOIV and AEFV, respectively. The SDU of all three frame positions of CpG in AEFV (Figure 3b) fall within the error distribution (SDU_CpGbridge_ = 0.96, error_MIN_ = 0.85, error_MAX_ = 1.22; SDU_CpGpos1_ = 1.08, error_MIN_ = 0.84, error_MAX_ = 1.15; SDU_CpGpos2_ = 0.87, error_MIN_ = 0.80, error_MAX_ = 1.22), whereas the respective SDU values in APOIV, the rodent infecting virus, (Figure 2b) are well below 1 (SDU_CpGbridge_ = 0.41, error_MIN_ = 0.82, error_MAX_ = 1.19; SDU_CpGpos1_ = 0.56, error_MIN_ = 0.82, error_MAX_ = 1.19; SDU_CpGpos2_ = 0.30, error_MIN_ = 0.79, error_MAX_ = 1.21), indicating significant CpG under-representation. These observations are in agreement with the previous research and the hypothesis of a CpG-targeting antiviral mechanism in vertebrates.

We also present the RDA and RSDU values for the two viruses, in order to make comparisons between the metrics (Figure 2a,c and Figure 3a,c). In contrast to the SDU, RDA values cannot be compared to a null distribution, so there is no statistical evaluation of over- and under-representation of dinucleotides. For example, in Figure 2 the CpU dinucleotide seems to be over-represented in all three frame positions according to the RDA metric. The SDU plot, however, shows that CpU bridge and position 1 values fall within the null distributions and only position 2 can be confidently described as over-represented (SDU_CpUpos2_ = 1.26). Thus, by using the SDU one can assess how confidently a value reflects over- or under-representation. For example, in Figure 3a, RDA suggests under-representation of position 2 and bridge of GpU and UpA (RDA_GpUbridge_ = 0.86, RDA_GpUpos2_ = 0.76, RDA_UpAbridge_ = 0.86, RDA_UpApos2_ = 0.71). According to our metric, the GpU values fall within the null distribution (SDU_GpUbridge_ = 1.02, error_MIN_ = 0.72, error_MAX_ = 1.28; SDU_GpUpos2_ = 0.96, error_MIN_ = 0.78, error_MAX_ = 1.22) and only UpA can be confidently called under-represented in the AEFV genome (SDU_UpAbridge_ = 0.75, error_MIN_ = 0.81, error_MAX_ = 1.24; SDU_UpApos2_ = 0.67, error_MIN_ = 0.84, error_MAX_ = 1.19) (Figure 3b).

The RSDU plots (Figure 2c and Figure 3c) showcase how the relative expected number of occurrences differs between dinucleotides and frame positions. For example, for both viruses UpC representation falls within the null distribution for all three frame positions, however, the expected RSDU value under the null hypothesis is much larger for UpC position 1 (Figure 2c and Figure 3c). This simply depends on the expected occurrences of codons (or codon pairs for bridge position) that contain this dinucleotide at that frame position under synonymous codon usage. The RSDU can be useful when comparing the level of over-representation between two dinucleotide combinations. The dinucleotides UpG and CpA are both over-represented at the bridge position in the APOIV genome, with identical SDU values (SDU_UpGbridge_ = 1.47, SDU_CpAbridge_ = 1.47). However, UpG is relatively more over-represented than CpA, as indicated by their RSDU values (RSDU_UpGbridge_ = 0.38, RSDU_CpAbridge_ = 0.23). The reason why the SDU values of the two are the same is because the maximum value SDU_UpGbridge_ can take (when all synonymous bridge positions have a UpG) is 3.33, much lower than the SDU_CpAbridge_ maximum value of 5.24.

The SDU of a given dinucleotide should directly reflect the RSCU of the codons that contain it (or in case of the bridge position: the first nucleotide in the third codon position and second nucleotide in the first codon position of the downstream amino acid). To illustrate this relation between the two metrics, we have calculated the RSCU for all codons of AEFV and APOIV (Table 2). ApG in position 1 seems to be over-represented in APOIV (SDU_ApGpos1_ = 1.51), which is not the case in AEFV (SDU_ApGpos1_ = 0.95). This is clearly depicted in the RSCU values of all AG-starting codons being higher in APOIV (Table 2). AGA is highly over-represented in the APOIV genome exclusively (APOIV: RSCU = 2.19; AEFV: RSCU = 1.00), which also explains the high SDU value for GpA in frame position 2 (APOIV: SDU_GpApos2_ = 1.63; AEFV: SDU_GpApos2_ = 0.96). This over-representation of AG-starting codons in APOIV also seems to drive the only clear inconsistency between SDU and RDA values, where position 1 ApG is under-represented based on RDA but over-represented based on SDU (Figure 2a,b). All CG-starting and -ending codons are under-represented in APOIV (Table 2), consistent with CpG depletion and low SDU_CpGpos1_ and SDU_CpGpos2_ in the vertebrate-infecting virus. While it is seemingly easy to trace such dinucleotide bias patterns in the 1st and 2nd frame position only by using the RSCU metric, this becomes much harder for the bridge position. For example, the SDU values for the bridge positions of ApG, GpA and CpG also fall outside the null hypothesis distribution in APOIV, but not in AEFV (Figure 2b and Figure 3b). This highlights the importance of having a metric for detecting compositional biases on the dinucleotide level.

### 3.3. SDU Shows Consistent CpG Differences between Insect- and Vertebrate-Specific Viruses

Since there is evidence for a vertebrate immune response selecting against CpG dinucleotides in viral genomes, we decided to further explore this trend between members of the *Flaviviridae* family, specific to and absent in vertebrate hosts using the SDU. First, we calculated the SDU of CpG for all frame positions for the two sets of insect-specific and vertebrate-specific (no known insect vector) viruses used by Simón et al. [25]. Figure 4a shows the SDU values for the two groups plotted against the total GC content of the viral coding sequences. As indicated by the graph, the CpG SDU correlates with the overall GC content with slopes that are not significantly different between hosts (GLM: F_1,57_ = 1.492, *p* = 0.227). However, the intercepts of the fitted lines are significantly different between the host groups with the host and the GC content predictors explaining 89% of the variance. The vertebrate-infecting viruses have significantly lower SDU values, thus more CpG depletion than invertebrate-specific flaviviruses (GLM: *R*^2^ = 0.89, F_2,57_ = 238.3, *p* < 0.0001).

This analysis was replicated on a set of viruses of the *Rhabdoviridae* family (Figure 4b). The trend observed is similar to that seen with the *Flaviviridae*. Both GC content and host-specificity explain 51% of the variance (GLM: *R*^2^ = 0.51, F_2,201_ = 105.4, *p* < 0.0001) around the CpG representation as quantified by our metric. The difference in rhabdovirus CpG depletion is not as large as in flaviviruses. CpG representation in invertebrate-infecting *Flaviviridae* is consistent with the null hypothesis of equal synonymous dinucleotides (mean SDU_CpG_ = 0.94), while vertebrate-infecting members of the family show a higher CpG suppression (mean SDU_CpG_ = 0.31). On the contrary, both sets of *Rhabdoviridae* have depleted CpG levels, vertebrate-specific ones having slightly more suppression (invertebrate-specific: mean SDU_CpG_ = 0.42, vertebrate-specific: mean SDU_CpG_ = 0.41).

## 4. Discussion

In this paper we propose SDU as a novel method for quantifying dinucleotide representation in a coding sequence by comparing the observed frequency of a synonymous dinucleotide to that expected under the null hypothesis of equal synonymous codon usage. We further extend this metric for comparisons between different dinucleotides with the RSDU formula and provide a means of measuring statistical error for the calculated values.

Using the coding sequences of two flaviviruses with different dinucleotide compositions, APOIV and AEFV, we present how the SDU compares to its extension, RSDU, and to the standard metric for dinucleotide quantification, RDA (Figure 2 and Figure 3). The RSDU can prove useful for comparing the level of over-representation between two dinucleotides, bypassing the variable scale of SDU values for different dinucleotide positions. Overall, there is agreement between the results of RDA and SDU. RDA values close to 1, however, can be misinterpreted as over- or under-representation when the value is not exactly equal to 1 just by chance. Our method provides a way of testing the significance of a value by comparing to a null distribution and accepting a skew in dinucleotide representation if the SDU value falls outside the expected distribution. This might be a more conservative way of assessing dinucleotide representation but avoids such misinterpretations.

We have explored the representation of different dinucleotides in a set of previously examined members of the *Flaviviridae* family, using the SDU, and produced results in agreement with previous research [25]. Based on suggestions from current research that a host-mediated antiviral response in vertebrates selects against CpGs in viral RNA, we focused on examining CpG representation in the *Flaviviridae* and a set of *Rhabdoviridae* viruses. In both datasets, the SDU calculations show a greater depletion of CpG in the vertebrate-specific viruses. Even though there is no experimental evidence that a CpG-recognising immune response is acting on flaviviruses or rhabdoviruses in vertebrates, other RNA viruses seem to be affected by such a mechanism [10,12,29]. Thus, we hypothesise that this immune mechanism is responsible for the difference in CpG SDU values between the two virus groups.

It is important to consider that factors other than a dinucleotide-targeting immune response are likely to bias viral genomic composition and such biases might also be reflected in the SDU values. For example, most invertebrate flaviviruses show weak or no CpG under-representation, while vertebrate-infecting flaviviruses show high CpG depletion. This pattern is less clear in the rhabdoviruses, where CpG is under-represented in both groups. We suggest this might be a result of other rhabdovirus-specific host mechanisms, or simply a lineage-specific mutational bias or the base composition of the *Rhabdoviridae*.

Finally, with the SDU, each informative frame position is examined separately for a given dinucleotide. This feature should be useful for exploring cases where selective or mutational processes are acting on specific positions in the coding sequence, or on longer motifs including the dinucleotide of interest. In comparison to the RDA, the SDU takes into account where changes due to a dinucleotide preference are more likely to take place, by assuming that synonymous nucleotide changes are more likely than non-synonymous ones. Thus, our metric reflects dinucleotide patterns, while being informed by a biological assumption. Interpretations of the metric should still be made with caution, since it does not disentangle between the forces affecting synonymous dinucleotide representation, whether that is selection or mutation on the dinucleotide level, or forces targeting codons and longer motifs. Whilst we have developed this metric to expand our understanding of dinucleotide biases in viruses, the SDU can have broader applications, for example in the host coding sequences.

## Figures and Tables

**Figure 1 viruses-12-00462-f001:**
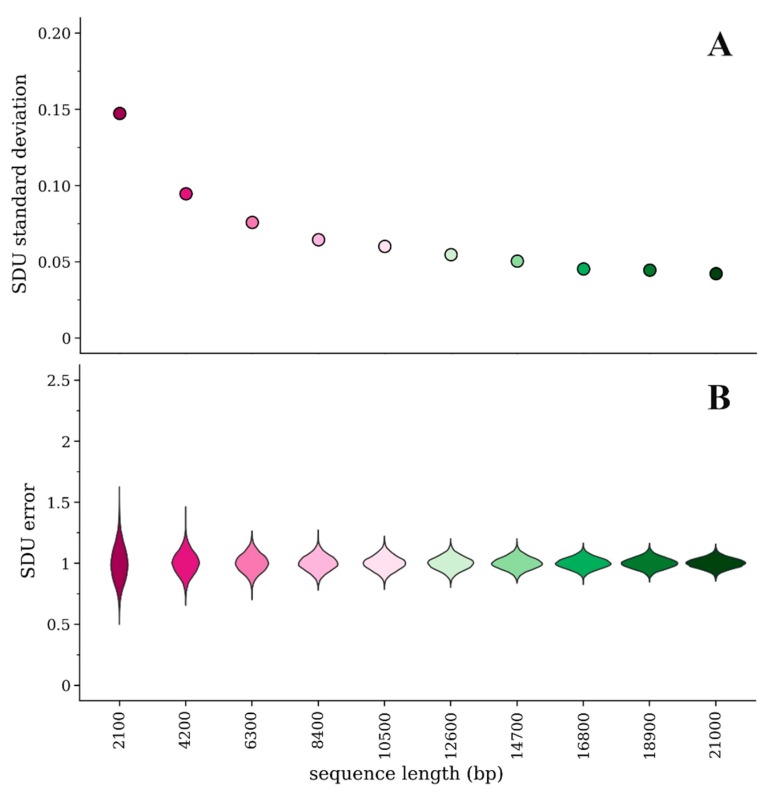
Comparison of error for the SDU_CpGbridge_ of 10 simulated amino acid sequences of different lengths with 1000 random samples of nucleotide sequences for each amino acid sequence: (**A**) Standard deviation of the mean of the SDU error distributions; (**B**) Violin plots of the error distribution for each simulated sequence.

**Figure 2 viruses-12-00462-f002:**
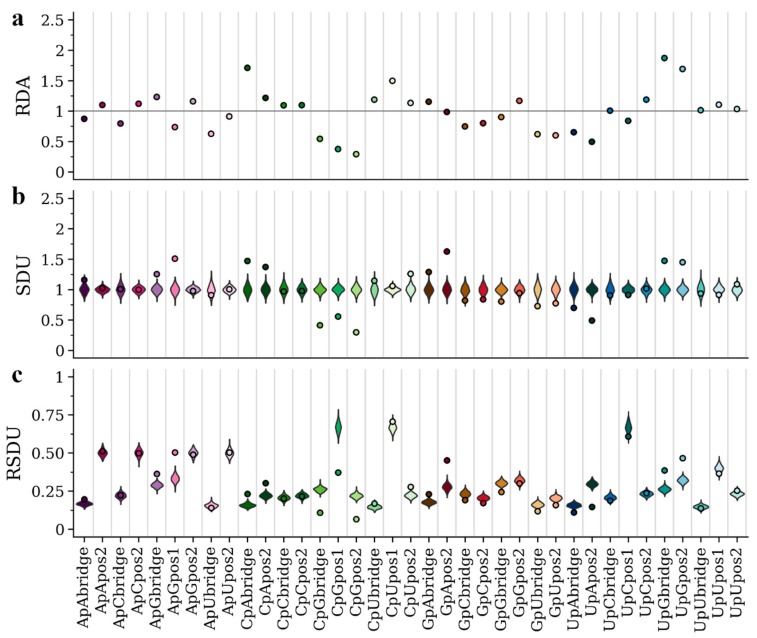
RDA (**a**), SDU (**b**) and RSDU (**c**) values for all informative dinucleotides and frame positions plotted for the APOIV coding sequence. Dot points indicate observed values and violin plots indicate SDU/RSDU error distributions around the null hypothesis (1000 random samples for each value). The grey horizontal line indicates an RDA of 1. Position 1 of dinucleotides CpC, CpA, GpC, GpG, GpU, GpA, UpG, UpA, ApC, ApU, ApA are excluded because they can only produce one amino acid (non-informative).

**Figure 3 viruses-12-00462-f003:**
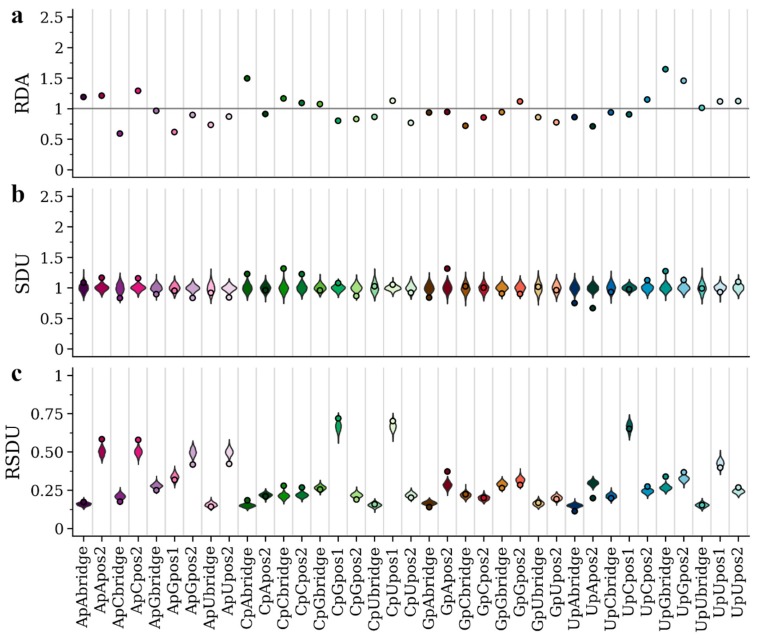
RDA (**a**), SDU (**b**) and RSDU (**c**) values for all informative dinucleotides and frame positions plotted for the AEFV coding sequence. Dot points indicate observed values and violin plots indicate SDU/RSDU error distributions around the null hypothesis (1000 random samples for each value). The grey horizontal line indicates an RDA of 1. Position 1 of dinucleotides CpC, CpA, GpC, GpG, GpU, GpA, UpG, UpA, ApC, ApU, ApA are excluded because they can only produce one amino acid (non-informative).

**Figure 4 viruses-12-00462-f004:**
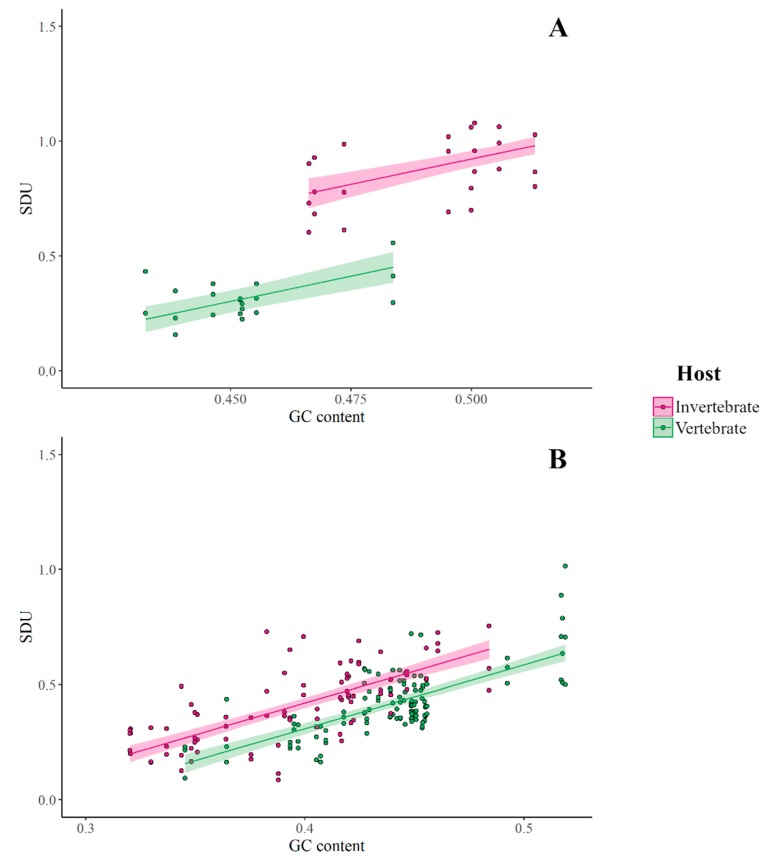
Comparison of SDU_CpG_ for all frame positions between invertebrate- and vertebrate-specific (**A**) *Flaviviridae* and (**B**) *Rhabdoviridae*, plotted against the overall GC content of the coding sequences (Appendix A).

**Table 1 viruses-12-00462-t001:** Notation used to define SDU and RSDU.

Symbol	Description
*i*	Amino acid or amino acid pair
*j*	Dinucleotide
*h*	Dinucleotide frame position
*n_i_*	Number of occurrences of amino acid or amino acid pair *i* in the sequence
*k*	Set of different amino acids or amino acid pairs present in the sequence
*o_i,j,h_*	Synonymous proportion of dinucleotide *j* in frame position *h* for amino acid or amino acid pair *i* observed in the sequence
*e_i,j,h_*	Synonymous proportion of dinucleotide *j* in frame position *h* for amino acid or amino acid pair *i* expected under equal synonymous codon usage
*N*	Total number of amino acids or amino acid pairs present in the sequence

**Table 2 viruses-12-00462-t002:** RSCU values for each codon, calculated for the APOIV and AEFV coding sequences. Highlighted in bold are the values mentioned in the text. No values have been calculated for stop codons (UAA, UAG, UGA), since only one coding sequence was used for each virus.

	APOIV	AEFV		APOIV	AEFV		APOIV	AEFV		APOIV	AEFV
**UUU**	1.11	1.08	**UCU**	1.01	0.75	**UAU**	0.98	1.00	**UGU**	0.95	1.04
**UUC**	0.89	0.92	**UCC**	0.73	1.13	**UAC**	1.02	1.00	**UGC**	1.05	0.96
**UUA**	0.38	0.56	**UCA**	1.57	0.96	**UAA**	STOP	STOP	**UGA**	STOP	STOP
**UUG**	1.39	1.23	**UCG**	**0.34**	**1.06**	**UAG**	STOP	STOP	**UGG**	1.00	1.00
**CUU**	1.08	0.83	**CCU**	1.19	0.89	**CAU**	1.25	1.05	**CGU**	**0.48**	**1.19**
**CUC**	1.00	1.42	**CCC**	0.86	1.00	**CAC**	0.75	0.95	**CGC**	**0.39**	**1.19**
**CUA**	0.67	0.79	**CCA**	1.69	1.32	**CAA**	0.91	1.23	**CGA**	**0.68**	**1.05**
**CUG**	1.48	1.17	**CCG**	**0.25**	**0.79**	**CAG**	1.09	0.77	**CGG**	**0.68**	**0.88**
**AUU**	1.10	1.22	**ACU**	1.13	0.98	**AAU**	0.91	0.84	**AGU**	**1.04**	**0.96**
**AUC**	1.15	1.05	**ACC**	1.27	1.07	**AAC**	1.09	1.16	**AGC**	**1.32**	**1.13**
**AUA**	0.75	0.73	**ACA**	1.29	1.00	**AAA**	1.01	1.23	**AGA**	**2.19**	**1.00**
**AUG**	1.00	1.00	**ACG**	**0.32**	**0.95**	**AAG**	0.99	0.77	**AGG**	**1.58**	**0.69**
**GUU**	1.08	1.29	**GCU**	1.65	1.07	**GAU**	0.97	0.67	**GGU**	0.73	0.74
**GUC**	1.00	0.98	**GCC**	1.03	1.62	**GAC**	1.03	1.33	**GGC**	0.73	0.73
**GUA**	0.29	0.62	**GCA**	1.05	0.68	**GAA**	1.09	1.07	**GGA**	1.75	1.57
**GUG**	1.63	1.11	**GCG**	**0.27**	**0.63**	**GAG**	0.91	0.93	**GGG**	0.80	0.96

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
