# Peer review of "Synonymous Dinucleotide Usage: A Codon-Aware Metric for Quantifying Dinucleotide Representation in Viruses"

_viruses, 2020, doi:10.3390/v12040462_

Round 1
Reviewer 1 Report
In this manuscript Lytas and Hughes develop a novel metric termed Synonymous Dinucleotide Usage (SDU) which links dinucleotide frequencies to amino acid codon usage frequencies in RNA viral genome sequences with the potential for application on RNA and DNA sequences of different origins. A software package (DinuQ) has been made available to the research community for the calculation of SDU values.
Minor comments:
The manuscript is clearly written and the findings are presented to a high standard. On occasion, I felt the language could have been simplified or explanations added to make the content more accessible.
Line 39/40: Decreasing CpG and UpA abundance in RNA viral genomes rarely leads to a significant increase in viral replication. Exceptions are altered transgenes integrated into viral replicons. The mechanism of this increased expression remains unclear and may not be linked to immunity.
Line 52: It is not clear what is meant by "pairs of codons" in this context
The main part of the results section of the manuscript describes the error associated to SDU calculations depending on input sequence length and the application of SDU and RSDU calculations to a previously analysed Flavivirus genome dataset. Given that comparisons between the two analyses were drawn, I would have liked to see examples of this comparison as part of Figure 2 and Table 2 to better understand how the newly developed SDU metric improves on "traditional" dinucleotide/codon usage analyses or what sort of additional biological information could be drawn from such adjusted analysis. Further, the biological novelty of the manuscript could be improved by adding an analysis that has not been done before or by presenting an example of where the new SDU/RSDU analyses lead to different conclusions on host adaptation etc. than previously published findings.
Supplementary table 1: Please add virus names rather than just accession numbers.
Author Response
We thank the reviewers for their feedback and suggestions, which we believe have helped improve our manuscript. We have taken all of their comments into considerations and have detailed the changes we have made below as well as provided a document with the changes highlighted (with track changes).
General statement
Lines 20-21: clarified sentence phrasing
Line 39: removed part of the sentence that was not entirely correct as pointed out by Reviewer 1
Line 51: minor rewording
Line 63-67: improved the introduction by providing additional references of existing metrics similar to ours
Lines 69-71: clarified sentence phrasing
Line 76: minor rewording
Line 100-102: explicitly stated that 11 position 1 dinucleotides have non-informative SDU values as pointed out by Reviewer 2
Line 175: updated the subheading to better match the reviewed content
Lines 180-185: reworded to match the updated figures
Lines 189-195: updated figure 2 and its legend to include RDA and RSDU plots for the first virus (APOIV)
Lines 196-209: improved the results section by making comparisons between our metric and the traditional RDA metric
Lines 211-217: updated figure 3 and its legend to include RDA and RSDU plots for the second virus (AEFV)
Lines 218-230: included a paragraph to demonstrate the usefulness of the RSDU metric
Lines 239-242: added a sentence relating to the SDU-RDA comparison
Lines 247-248, 259. 267, 269: updated figure annotations
Line 256: explicitly stated the virus family under investigation as pointed out by Reviewer 2
Line 264: minor rewording
Lines 284-293: improved the discussion by including a paragraph on the comparison between the metrics
Lines 294-295: minor rewording
Lines 311-320: updated the last paragraph to clarify how our metric compares to the RDA
Supplementary Data: added the virus names to Table S1 as requested by Reviewer 1
Reviewer 1
In this manuscript Lytras and Hughes develop a novel metric termed Synonymous Dinucleotide Usage (SDU) which links dinucleotide frequencies to amino acid codon usage frequencies in RNA viral genome sequences with the potential for application on RNA and DNA sequences of different origins. A software package (DinuQ) has been made available to the research community for the calculation of SDU values.
Minor comments:
The manuscript is clearly written and the findings are presented to a high standard. On occasion, I felt the language could have been simplified or explanations added to make the content more accessible.
Line 39/40: Decreasing CpG and UpA abundance in RNA viral genomes rarely leads to a significant increase in viral replication. Exceptions are altered transgenes integrated into viral replicons. The mechanism of this increased expression remains unclear and may not be linked to immunity.
- We have reworded the sentence on line 39.
Line 52: It is not clear what is meant by "pairs of codons" in this context
- We have reworded the sentence now on line 51.
The main part of the results section of the manuscript describes the error associated to SDU calculations depending on input sequence length and the application of SDU and RSDU calculations to a previously analysed Flavivirus genome dataset. Given that comparisons between the two analyses were drawn, I would have liked to see examples of this comparison as part of Figure 2 and Table 2 to better understand how the newly developed SDU metric improves on "traditional" dinucleotide/codon usage analyses or what sort of additional biological information could be drawn from such adjusted analysis. Further, the biological novelty of the manuscript could be improved by adding an analysis that has not been done before or by presenting an example of where the new SDU/RSDU analyses lead to different conclusions on host adaptation etc. than previously published findings.
- We have now detailed examples where RDA would suggest over-representation (CpU) but put into context of the null distribution, the SDU does not support this. Also, cases where the RDA does not suggest any under or over-representation (GpA position 1) but the comparison of the SDU to the null distribution would suggest over-representation. We have added the RDA and SDU to figures 2 & 3, which provides a comparison of the different metrics.
Supplementary table 1: Please add virus names rather than just accession numbers.
- We have added a ‘description’ column with the names of the viruses to the supplementary table.
Reviewer 2 Report
The relative frequency of neighboring nucleotides (dinucleotides) is a relatively stable feature of the genome of an organism. It is known that vertebrates have strong suppression of CpG and moderate suppression of TpA dinucleotides in their DNA genomes. The CpG suppression in organisms with methylated genome is caused by the spontaneous deamination of methylcytosine, whereby cytosine mutates to thymine. The most common explanation for widespread TpA (UpA) suppression is that UpA dinucleotides destabilize mRNA molecules because UpA-rich regions are efficiently digested by cellular RNases. However, it remains relatively unexplained why the genomes of many RNA viruses that infect vertebrates also show suppression of CpG and UpA dinucleotides.
The authors of this study developed the synonymous dinucleotide use metrics (SDU) to better understand the nature of dinucleotide variations in nucleic acid sequences. The SDU compares the occurrence of a given dinucleotide with the null hypothesis that there is an equal use of synonymous codons. The calculation of the SDU is relatively straightforward. The SDU can be calculated for each of the 16 possible dinucleotides and 3 different codon pair positions.
The main concerns
As the authors mention, the deviation of dinucleotide frequencies from the mathematical prediction is usually evaluated by calculating the relative dinucleotide frequency (RDA). The RDA is the ratio of the observed dinucleotide frequencies to the expected dinucleotide frequencies in the analyzed sequence. Although theoretically useful, the authors have not given examples of advantages of the SDU over the current standard, the RDA.
The authors claim that the SDU metric provides a better estimate of dinucleotide deviations from predicted values than the RDA, but the authors do not show a comparison of the two metrics for analyzed virus sequences in the manuscript. This is surprising because the Python package developed by the authors also contains a module for calculating the RDA. For example, the authors could compare the performance of the SDU that of the RDA in Figure 2.
The authors assert that the SDU allows a biologically relevant interpretation of SDU values. Could the authors provide an example of such a biologically relevant interpretation that they have discovered in this study and that was not possible via the RDA?
The RDA, similar to the SDU, can be calculated for each of the 16 possible dinucleotides and 3 different codon positions. Since only three of twenty available amino acids are encoded by codons starting with dissimilar dinucleotides (arginine, leucine and serine), the SDU calculated for most dinucleotides at codon position 1 is not informative (11/16). Therefore, the RDA at codon position 1 seems to be more sensitive to the suppression of dinucleotides than the SDU.
The authors describe an extension of SDU in the form of a relative SDU (RSDU) metrics, but they do not show an example that would demonstrate the usefulness of this calculation in the main text.
Minor issues
In Paragraph 3.3 the authors fail to inform what virus genomes are analyzed. First the last sentence of this paragraph mentions that the analysis describes situation in flaviviruses (line 216).
Author Response
We thank the reviewers for their feedback and suggestions, which we believe have helped improve our manuscript. We have taken all of their comments into considerations and have detailed the changes we have made below as well as provided a document with the changes highlighted (with track changes).
General statement
Lines 20-21: clarified sentence phrasing
Line 39: removed part of the sentence that was not entirely correct as pointed out by Reviewer 1
Line 51: minor rewording
Line 63-67: improved the introduction by providing additional references of existing metrics similar to ours
Lines 69-71: clarified sentence phrasing
Line 76: minor rewording
Line 100-102: explicitly stated that 11 position 1 dinucleotides have non-informative SDU values as pointed out by Reviewer 2
Line 175: updated the subheading to better match the reviewed content
Lines 180-185: reworded to match the updated figures
Lines 189-195: updated figure 2 and its legend to include RDA and RSDU plots for the first virus (APOIV)
Lines 196-209: improved the results section by making comparisons between our metric and the traditional RDA metric
Lines 211-217: updated figure 3 and its legend to include RDA and RSDU plots for the second virus (AEFV)
Lines 218-230: included a paragraph to demonstrate the usefulness of the RSDU metric
Lines 239-242: added a sentence relating to the SDU-RDA comparison
Lines 247-248, 259. 267, 269: updated figure annotations
Line 256: explicitly stated the virus family under investigation as pointed out by Reviewer 2
Line 264: minor rewording
Lines 284-293: improved the discussion by including a paragraph on the comparison between the metrics
Lines 294-295: minor rewording
Lines 311-320: updated the last paragraph to clarify how our metric compares to the RDA
Supplementary Data: added the virus names to Table S1 as requested by Reviewer 1
Reviewer 2
The relative frequency of neighboring nucleotides (dinucleotides) is a relatively stable feature of the genome of an organism. It is known that vertebrates have strong suppression of CpG and moderate suppression of TpA dinucleotides in their DNA genomes. The CpG suppression in organisms with methylated genome is caused by the spontaneous deamination of methylcytosine, whereby cytosine mutates to thymine. The most common explanation for widespread TpA (UpA) suppression is that UpA dinucleotides destabilize mRNA molecules because UpA-rich regions are efficiently digested by cellular RNases. However, it remains relatively unexplained why the genomes of many RNA viruses that infect vertebrates also show suppression of CpG and UpA dinucleotides.
The authors of this study developed the synonymous dinucleotide use metrics (SDU) to better understand the nature of dinucleotide variations in nucleic acid sequences. The SDU compares the occurrence of a given dinucleotide with the null hypothesis that there is an equal use of synonymous codons. The calculation of the SDU is relatively straightforward. The SDU can be calculated for each of the 16 possible dinucleotides and 3 different codon pair positions.
The main concerns
As the authors mention, the deviation of dinucleotide frequencies from the mathematical prediction is usually evaluated by calculating the relative dinucleotide frequency (RDA). The RDA is the ratio of the observed dinucleotide frequencies to the expected dinucleotide frequencies in the analyzed sequence. Although theoretically useful, the authors have not given examples of advantages of the SDU over the current standard, the RDA.
The authors claim that the SDU metric provides a better estimate of dinucleotide deviations from predicted values than the RDA, but the authors do not show a comparison of the two metrics for analyzed virus sequences in the manuscript. This is surprising because the Python package developed by the authors also contains a module for calculating the RDA. For example, the authors could compare the performance of the SDU that of the RDA in Figure 2.
- We have made comparisons between the RDA and the SDU in lines 196-209 and 239-242 and added plots of the RSDU and RDA (figures 2 & 3). We further address the points above in the Discussion, in lines 289-294.
The authors assert that the SDU allows a biologically relevant interpretation of SDU values. Could the authors provide an example of such a biologically relevant interpretation that they have discovered in this study and that was not possible via the RDA?
We have clarified this point in lines 20-21, 70-72 and 315-318.
The RDA, similar to the SDU, can be calculated for each of the 16 possible dinucleotides and 3 different codon positions. Since only three of twenty available amino acids are encoded by codons starting with dissimilar dinucleotides (arginine, leucine and serine), the SDU calculated for most dinucleotides at codon position 1 is not informative (11/16). Therefore, the RDA at codon position 1 seems to be more sensitive to the suppression of dinucleotides than the SDU.
- We explicitly state this limitation of the SDU in lines 100-102.
The authors describe an extension of SDU in the form of a relative SDU (RSDU) metrics, but they do not show an example that would demonstrate the usefulness of this calculation in the main text.
- We have demonstrated the usefulness of the RSDU in lines 218-230 and have included RSDU plots in figures 2c and 3c.
Minor issues
In Paragraph 3.3 the authors fail to inform what virus genomes are analyzed. First the last sentence of this paragraph mentions that the analysis describes situation in flaviviruses (line 216).
- We have clarified this point in line 257.